# Awareness of Malocclusion Is Closely Associated with Allergic Rhinitis, Asthma, and Arrhythmia in Late Adolescents

**DOI:** 10.3390/healthcare8030209

**Published:** 2020-07-12

**Authors:** Masanobu Abe, Akihisa Mitani, Atsushi Yao, Liang Zong, Kazuto Hoshi, Shintaro Yanagimoto

**Affiliations:** 1Division for Health Service Promotion, The University of Tokyo, Tokyo 113-0033, Japan; mitania-int@h.u-tokyo.ac.jp (A.M.); yaoa-int@h.u-tokyo.ac.jp (A.Y.); yanagimoto@hc.u-tokyo.ac.jp (S.Y.); 2Department of Oral & Maxillofacial Surgery, The University of Tokyo Hospital, Tokyo 113-8655, Japan; hoshi-ora@h.u-tokyo.ac.jp; 3Division of Epigenomics, National Cancer Center Research Institute, Tokyo 104-0045, Japan; zl20014111@163.com

**Keywords:** awareness of malocclusion, malocclusion, occlusal disorder, adolescence, systemic disease, asthma, allergic rhinitis, arrhythmia, heart disease

## Abstract

Background: Oral infectious diseases are typified by dental caries and periodontal diseases and are known to be associated with various systemic diseases. However, clear associations between occlusal disorders and systemic diseases have not yet been established. We investigated the association between an awareness of malocclusion and common diseases in late adolescence. Methods: We retrospectively reviewed the data of mandatory medical questionnaires that are a legal requirement of the freshman medical checkup. We collected the data of all the students who completed the questionnaires between April 2017 and 2019. The data were analyzed using the *χ^2^* test and multivariate analysis was performed with a binomial logistic regression model. Results: The subjects were 9098 students aged 17–19 (mean age 18.3 years). The rate of awareness of malocclusion was 2.14% (195 out of 9098 eligible subjects; 160 males and 35 females). These students had significantly greater rates of allergic rhinitis, asthma, and arrhythmia (*p* < 0.001). Multivariate analysis revealed significant rates of allergic rhinitis (odds ratio [OR] 2.184, 95% confidence interval [CI]: 1.468–3.250, *p* < 0.001), asthma (OR 1.843, 95%CI: 1.153–2.945, *p* = 0.011), and arrhythmia (OR 2.809, 95%CI: 1.083–7.288, *p* = 0.034) among students with an awareness of malocclusion. Conclusion: We identified close associations between an awareness of malocclusion and systemic diseases including allergic rhinitis, asthma, and arrhythmia in the late adolescent population. These results reinforce the associations between malocclusion and allergic rhinitis and asthma, as well as providing novel insight into the association of malocclusion and arrhythmia. However, further research is necessary to confirm the associations and explore the mechanisms underlying these associations.

## 1. Background

Oral infectious diseases that are typified by dental caries and periodontal diseases are associated with various systemic diseases including heart disease, diabetes, respiratory disease, rheumatism, metabolic syndrome, systemic infection, and malignant tumors [1,2,3,4,5,6,7,8,9,10]. Our recent study revealed new insights into the close association of gum bleeding with otitis media/externa in late adolescence [11]. However, the association between occlusal disorders and systemic disease has not yet been established. The present study focused on the association between an awareness of malocclusion and common diseases during late adolescence.

Upper and lower airway obstruction has been reported to be related to malocclusion in adolescents [12,13,14]. The possible causes of upper airway obstruction include allergic rhinitis, hypertrophied adenoids and tonsils, congenital nasal deformities, and polyps. These conditions can lead to malocclusion as a consequence of mouth breathing due to nasal obstruction [12,13]. Asthma has been indicated to be a cause of lower airway obstruction associated with malocclusion in adolescents [14]. The mechanism of this association has not been fully elucidated, but it has been suggested that changes in muscular function create an environment that results in the development of dentofacial anomalies [15]. 

We conducted the present study to investigate the relationship between an awareness of malocclusion and common diseases during late adolescence. The identification of any such associations will contribute both to the treatment of existing disease, as well as to the prevention of diseases that may emerge at a later stage.

## 2. Methods

### 2.1. Study Design and Population

Mandatory medical questionnaires are a legal requirement of freshman medical checkups; we retrospectively reviewed the data of all questionnaires completed between April 2017 and 2019. The questionnaire is self-administered and consists of closed-ended and open-ended questions. In total, the questionnaire was distributed to 9376 students during the specified period. 

### 2.2. Questionnaire

The questionnaire was distributed to the students prior to the beginning of the medical checkups. The presence of an awareness of malocclusion was assessed by the yes/no question, “Do you have any difficulties chewing food due to malalignment of your teeth?” In the case of a “Yes” answer, the subject was categorized as having an awareness of malocclusion. The subject’s medical history was assessed by the question, “Please check off the presence or absence of the following medical histories: neuromuscular disease; cerebrovascular disease; coronary artery disease; other vascular disease; heart disease, arrhythmia, or abnormal electrocardiogram (ECG); hypertension; respiratory disease; esophageal or gastrointestinal disease; liver, gall bladder, or pancreatic disease; renal disease, abnormal data in a urinalysis; hyperuricemia, gout; diabetes mellitus; dyslipidemia; thyroid disease; endocrine disease; collagen disease; blood disease; malignant neoplasm; allergic disease; eye disease; otorhinolaryngologic disease; skin disease; bone, joint or muscle disease; urological disease or gynecological disease; and other”.

When a history of disease was reported, the subject was asked to specify the name(s) of the disease(s) (Appendix A). After evaluating the responses of each subject, we analyzed the associations between an awareness of malocclusion due to teeth malalignment and diseases/disorders. Acute diseases/disorders and relatively rare diseases/disorders (i.e., those identified in <50 subjects) were excluded from the analysis.

### 2.3. Statistical Analysis

The data were analyzed using the *χ^2^* test. We performed multivariate analysis with the use of a binomial logistic regression model. A *p*-value of <0.05 (two-sided) was accepted as statistically significant. We used the statistical software program SAS ver. 9.4 (SAS Institute Inc., Cary, NC, USA) for all analysis.

### 2.4. Ethical Approval

This study was approved by the research ethics committee of the University of Tokyo in 2018, approval no. 18-197 (currently, revised as no. 19-324 in 2019).

## 3. Results

### 3.1. Rate of Awareness of Malocclusion

We analyzed data from 9098 (aged less than 20 years) of the 9376 who received the questionnaire. The subjects were aged 17–19 years (mean age: 18.3 years) and included 7316 males and 1782 females who entered the University of Tokyo between April 2017 and 2019. The rate of awareness of malocclusion was higher among males than females (Table 1), although this was not statistically significant (*p* = 0.623).

### 3.2. Prevalence of Common Systemic Diseases/Disorders in Adolescence

The rates of common systemic diseases among the subjects are detailed in Table 2. The prevalence of pollinosis, food/drug allergy, inhaled antigen allergy (except pollinosis), allergic rhinitis, otitis media/externa, sinusitis, pneumothorax/mediastinal emphysema, asthma/cough-variant asthma, and atopic dermatitis were reported in our previous study [11]. In the current study, we identified the prevalence of urticaria, scoliosis, spondylosis/spondylolisthesis/hernia, strabismus, myopia/hyperopia/astigmatism, arrhythmia, abnormal ECG other than arrhythmia, and anemia to be 1.24%, 0.73%, 0.60%, 0.75%, 0.92%, 1.18%, 0.76%, and 0.67%, respectively. Within the category of arrhythmia, the numbers of subjects with premature cardiac complex, tachyarrhythmia, bradyarrhythmia, detail-unknown arrhythmia, and Brugada syndrome were 33, 18, 19, 36, and 1, respectively. Tachyarrhythmia included Wolff–Parkinson–White (WPW) syndrome (*n* = 14) and supraventricular tachycardia (*n* = 4). Bradyarrhythmia included high-grade atrioventricular block (*n* = 16), junctional rhythm (*n* = 1), and detail-unknown bradycardia (*n* = 2). 

### 3.3. Associations Between Awareness of Malocclusion and Common Systemic Diseases/Disorders in Adolescence

As Table 2 shows, an awareness of malocclusion was associated with a significantly greater incidence of a history of allergic rhinitis, asthma/cough-variant asthma, and arrhythmia. No significant associations with any other diseases/disorders were observed.

The incidence of a history of allergic rhinitis, asthma/cough-variant asthma, and arrhythmia was significantly associated with the presence of an awareness of malocclusion among the male subjects, while spondylosis/spondylolisthesis/hernia was significantly associated with an awareness of malocclusion among the female subjects.

Multivariate analysis using a binomial logistic regression model with an awareness of malocclusion as the objective variable (awareness of malocclusion as an event) and the 17 above-mentioned diseases/disorders (pollinosis, food/drug allergy, inhaled antigen allergy, allergic rhinitis, otitis media/externa, sinusitis, pneumothorax/mediastinal emphysema, asthma/cough-variant asthma, atopic dermatitis, urticaria, scoliosis, spondylosis/spondylolisthesis/hernia, strabismus, myopia/hyperopia/astigmatism, arrhythmia, abnormal ECG other than arrhythmia, and anemia) plus gender male and gum bleeding [11,16] as explanatory variables revealed an awareness of malocclusion to be closely associated with allergic rhinitis, asthma/cough-variant asthma, and arrhythmia (Table 3).

## 4. Discussion

Dental caries and periodontal disease are common infectious oral diseases, and are known to be associated with various systemic diseases [1,2,3,4,5,6,7,8,9,10]. We recently reinforced the new evidence of the association between periodontal disease and asthma, as well as novel insight into the close association between gingivitis and otitis media/externa [11]. In the present study, we focused on malocclusion in adolescence, and our analysis of over 9000 university students identified close associations between the awareness of malocclusion and allergic rhinitis, asthma, and arrhythmia. These results reinforce the association between malocclusion and allergic rhinitis and asthma, and demonstrate a new association between malocclusion and arrhythmia.

We found an awareness of malocclusion to be closely and significantly associated with a history of allergic rhinitis in the present adolescent population. Multivariate analysis identified allergic rhinitis as an independent factor for an awareness of malocclusion. Mouth breathing due to upper airway obstruction (nasal obstruction) is likely to have negative effects on the general development of the cranial complex. Furthermore, mouth breathing is associated with the development of dental malocclusion including anterior open bite and large overjet [12,13,14].

Allergic rhinitis is the most common upper airway-obstructing disease to be experienced in adolescence [17,18,19], and is thus likely to cause malocclusion due to mouth breathing [14,20,21]. Allergic rhinitis has also been reported to be a risk factor for dental trauma [22]. Other possible causes of mouth breathing include hypertrophied adenoids and tonsils, congenital nasal deformities, and polyps [12,13], but the incidences of these conditions in the present study were too small to analyze their association with the awareness of malocclusion.

Our findings reveal an awareness of malocclusion to be closely and significantly associated with a history of asthma in late adolescence, which was found to be an independent factor for an awareness of malocclusion. Previous investigations indicated that adolescents with asthma are more likely to have malocclusions, particularly anterior open bite, compared with healthy controls (OR 1.78, *p* = 0.017), in line with the results of the present study [14]. Although the mechanism underlying the association between asthma and malocclusion remains unclear, it has been suggested that changes in muscular function are involved [15]. 

Notably, we identified an association between an awareness of malocclusion and arrhythmia and determined this condition to be an independent factor for an awareness of malocclusion. To date, relationships between malocclusion and heart diseases have not been investigated. It has been reported that malocclusion, as chronic stress, contributes to reduced heart rate variability (HRV) in healthy young adults [23]. The HRV is known to be one of the most reliable methods for assessing autonomic activity [24]. Thus, malocclusion is likely to induce arrhythmia by increasing sympathetic nerve activity. Changes in vertical occlusal dimension have also been shown to affect heart rate fluctuations in guinea pigs [25]. On the basis of the results and the present and previous studies, it is possible that orthodontic treatment could contribute to the treatment and prevention of arrhythmia and other heart diseases. 

We identified the close associations between an awareness of malocclusion and systemic diseases in late adolescents. Further research is necessary to confirm the associations in late adolescents and to investigate whether the current results are common in all ages or not.

## 5. Conclusions

Our analysis of the data of over 9000 university students demonstrates the close association between an awareness of malocclusion and allergic rhinitis, asthma, and arrhythmia. These findings confirm previous suggestions of the association between malocclusion and allergic rhinitis and asthma, and provide novel insights into the close association between malocclusion and arrhythmia in adolescence. Further research is warranted to confirm the associations that were found in this study and explore the mechanisms underlying these associations, but our results indicate that early dental or orthodontic visits in adolescents might enable the early detection and timely treatment of malocclusion as well as the prevention and improvement of systemic diseases.

## Figures and Tables

**Table 1 healthcare-08-00209-t001:** The rate of an awareness of malocclusion among the study population.

		Total (*n* = 9098)		Male (*n* = 7316)		Female (*n* = 1782)	
		*n*	%	*n*	%	*n*	%
Awareness of malocclusion	Presence	195	2.14	160	2.19	35	1.96
	Absence	8903	97.86	7156	97.81	1747	98.04

**Table 2 healthcare-08-00209-t002:** Results of analyses of the associations between an awareness of malocclusion and common diseases in adolescence.

		All			Male			Female		
		Awareness of Malocclusion		*p-Value*	Awareness of Malocclusion		*p-Value*	Awareness of Malocclusion		*p-Value*
		*n* (%)			*n* (%)			*n* (%)		
		Presence	Absence		Presence	Absence		Presence	Absence	
Medical history	*n*	195 (100)	8903 (100)	-	160 (100)	7156 (100)	-	35 (100)	1747 (100)	-
Pollinosis	1417	25 (12.82)	1392 (15.64)	0.331	19 (11.88)	1107 (15.47)	0.256	6 (17.14)	285 (16.31)	1
Food/Drug allergy	276	10 (5.13)	266 (2.99)	0.130	9 (5.63)	217 (3.03)	0.100	1 (2.86)	49 (2.80)	1
Inhaled antigen allergy (except pollinosis)	184	6 (3.08)	178 (2.00)	0.424	6 (3.75)	140 (1.96)	0.187	0 (0.00)	38 (2.18)	0.771
Allergic rhinitis	1372	50 (25.64)	1322 (14.85)	<0.001 *	47 (29.38)	1101 (15.39)	<0.001 *	3 (8.57)	221 (12.65)	0.643
Otitis media/externa	180	4 (2.05)	176 (1.98)	1	3 (1.88)	146 (2.04)	1	1 (2.86)	30 (1.72)	1
Sinusitis	145	6 (3.08)	139 (1.56)	0.167	6 (3.75)	116 (1.62)	0.077	0 (0.00)	23 (1.32)	1
Pneumothorax/Mediastinal emphysema	111	1 (0.51)	110 (1.24)	0.562	1 (0.63)	106 (1.48)	0.576	0 (0.00)	4 (0.23)	1
Asthma/Cough variant asthma	873	34 (17.44)	839 (9.42)	<0.001 *	31 (19.38)	717 (10.02)	<0.001 *	3 (8.57)	122 (6.98)	0.976
Atopic dermatitis	640	15 (7.69)	625 (7.02)	0.825	13 (8.13)	516 (7.21)	0.774	2 (5.71)	109 (6.24)	1
Urticaria	113	3 (1.54)	110 (1.24)	0.959	2 (1.25)	90 (1.26)	1	1 (2.86)	20 (1.14)	0.89
Scoliosis	66	3 (1.54)	63 (0.71)	0.355	1 (0.63)	29 (0.41)	1	2 (5.71)	34 (1.95)	0.336
Spondylosis/Spondylolisthesis/Hernia	55	3 (1.54)	52 (0.58)	0.217	1 (0.63)	44 (0.61)	1	2 (5.71)	8 (0.46)	0.003 *
Strabismus	68	1 (0.51)	67 (0.75)	1	1 (0.63)	48 (0.67)	1	0 (0.00)	19 (1.09)	1
Myopia/Hyperopia/Astigmatism	84	2 (1.03)	82 (0.92)	1	2 (1.25)	64 (0.89)	0.962	0 (0.00)	18 (1.03)	1
Arrhythmia	107	8 (4.10)	99 (1.11)	<0.001 *	7 (4.38)	80 (1.12)	<0.001 *	1 (2.86)	19 (1.09)	0.862
Abnormal ECG other than arrhythmia	69	3 (1.54)	66 (0.74)	0.394	3 (1.88)	61 (0.85)	0.345	0 (0.00)	5 (0.29)	1
Anemia	61	22 (0.66)	39 (0.68)	1	11 (0.40)	20 (0.44)	0.917	11 (2.03)	19 (1.53)	0.577

Abbreviations: ECG, electrocardiogram. Notes: * *p* < 0.05.

**Table 3 healthcare-08-00209-t003:** Results of multivariate analyses of the associations between an awareness of malocclusion and common systemic diseases in adolescence.

Medical History	Odds Ratio (95% Confidence Interval)	*p-Value*
Pollinosis	0.712 (0.418–1.212)	0.211
Food/Drug allergy	1.589 (0.719–3.511)	0.252
Inhaled antigen allergy (except pollinosis)	1.682 (0.648–4.362)	0.285
Allergic rhinitis	2.184 (1.468–3.250)	<0.001 *
Otitis media/externa	0.704 (0.171–2.901)	0.627
Sinusitis	1.663 (0.590–4.684)	0.336
Pneumothorax/Mediastinal emphysema	0.883 (0.120–6.499)	0.903
Asthma/Cough variant asthma	1.843 (1.153–2.945)	0.011*
Atopic dermatitis	0.707 (0.348–1.436)	0.338
Urticaria	0.468 (0.063–3.461)	0.457
Scoliosis	1.130 (0.152–8.420)	0.905
Spondylosis/Spondylolisthesis/Hernia	1.099 (0.148–8.160)	0.926
Strabismus	0.758 (0.101–5.669)	0.787
Myopia/Hyperopia/Astigmatism	0.678 (0.088–5.209)	0.708
Arrhythmia	2.809 (1.083–7.288)	0.034 *
Abnormal ECG other than arrhythmia	2.174 (0.650–7.270)	0.208
Anemia	1.044 (0.139–7.862)	0.967
Gender male	1.032 (0.650–1.639)	0.894
Gum bleeding	1.369 (0.962–1.949)	0.081

Notes: * *p* < 0.05. Abbreviations: ECG, electrocardiogram.

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
