# Peer review of "Awareness of Malocclusion Is Closely Associated with Allergic Rhinitis, Asthma, and Arrhythmia in Late Adolescents"

_healthcare, 2020, doi:10.3390/healthcare8030209_

Round 1

Reviewer 1 Report

The manuscript submitted to Healthcare entitled “Awareness of Malocclusion Is Closely Associated with Allergic Rhinitis, Asthma, and Arrhythmia in Late Adolescents” is an original article which aim to investigate possible relation between awareness of malocclusion and common diseases in adolescence.

On my opinion the article is interesting, well written, with good English. The content of the manuscript is very interesting. The authors showed relationship between malocclusion and allergic rhinitis and asthma, as well as providing novel insight into the association of malocclusion and arrhythmia in a student population of Japan.

However, I highlighted some issues.

Introduction. Are there studies in the literature concerning diseases in the student population? Better specify the objectives and methods of the study.

Methods. Is it possible to insert a copy of the questionnaire as supplementary material? Specify the year of approval of the ethics committee. Is it a university or hospital ethics committee?

Results. Have students been diagnosed with malocclusion?

Discussion. Are there other similar studies that have shown similar results in the adult population? Did the authors find limitations in their study by comparing it with other in the literature?

Conclusion. Please improve. The interpretation of the results is not clear. No study limits are mentioned.

Author Response

We appreciate this reviewer's precise and constructive comments.

Comments:

  1. Are there studies in the literature concerning diseases in the student population? Better specify the objectives and methods of the study.

We thank the reviewer for this comment. The objectives and methods were specified owing to the comment.

  1. Is it possible to insert a copy of the questionnaire as supplementary material?

We prepared an additional file  (Supplementary File 1) for the questionnaire.

  1. Specify the year of approval of the ethics committee. Is it a university or hospital ethics committee?

This study was approved by the research ethics committee of the University of Tokyo in 2018, approval no. 18-197. The approval was revised as no. 19-324 in 2019. This description was added to Method section.

  1. Have students been diagnosed with malocclusion?

As the reviewer pointed out, the students have not been diagnosed as malocclusion. Thus, we carefully checked our manuscript and used the word "awareness" of malocculusion through the whole.

  1. Are there other similar studies that have shown similar results in the adult population? Did the authors find limitations in their study by comparing it with other in the literature?

We thank this reviewer's critical comment. No similar studies could be found in adult population. We described the limitations of our study in the Discussion section.

  1. Please improve. The interpretation of the results is not clear. No study limits are mentioned.

Owing to this comment, Conclusion was modified and study limits were mentioned.

Reviewer 2 Report

The prevalence rate of allergic rhinitis, asthma/cough variant asthma and arrythmia is significantly higher in only male adolescents with malocclusion not in female adolescents with malocclusion at Table 2. However, the prediction model of Table 3 were represented in only total adolescents. So another model with the adjustment for sex should be needed at Table 3.

At p. 2, line 85 of the manuscript, '3.1. Rate of Areness of Malocclusion' seems to be a typing error. 

Author Response

Comments:

  1. The prevalence rate of allergic rhinitis, asthma/cough variant asthma and arrythmia is significantly higher in only male adolescents with malocclusion not in female adolescents with malocclusion at Table 2. However, the prediction model of Table 3 were represented in only total adolescents. So another model with the adjustment for sex should be needed at Table 3.

We appreciate this reviewer's comment. The number of incidence (awareness of malocclusion) in female is too small to perform multivariate analysis, thus we have not prepared the model with adjustment for gender.  In stead of it, we put "gender male" as an explanatory variable in the multivariate analysis.

  1. At p. 2, line 85 of the manuscript, '3.1. Rate of Areness of Malocclusion' seems to be a typing error. 

We thank for this comment. We corrected the typing error.

Round 2

Reviewer 2 Report

Reviewer's opinions were reflected at the revised manuscript. If you do further study on this thesis, the analysis by sex is recommended because the prevalence rate  of allergic rhinitis, asthma and arrhythmia is significantly different. The  malocclusion may be not likely associated with rhinitis, asthma and arrythmia in late female adoscents.